# An Empirical Study of the Impact of Social Media Use on Online Political Participation of University Students in Western China

**Yulong Tang [1] and Qing Wen [2],***

[1] School of Journalism and Communication, Beijing Institute of Graphic Communication, Beijing 102699, China
[2] Institute of Communication, Communication University of China, Beijing 100024, China
* Correspondence: wenqing1004@cuc.edu.cn

**Abstract:** More and more university students in China are opting to access, share, and comment on political issues via social media as a result of the rapid expansion of Internet technology. In the western part of China, you can find Western universities. University students there find it challenging to use the Internet and engage in online political activities due to the region's level of economic development and social conventions. We are unsure whether their political involvement will have an effect on how society functions as a result. This study uses 530 students from Western colleges as a sample to investigate the effects of social media use on online political involvement and the adjustment effect of political efficacy. It combines a correlation analysis, multiple regression analysis, and regression multiples with a questionnaire survey. The results showed a strong positive link between Western Chinese university students' online political participation and their use of social media. Students' online political engagement in Western colleges greatly increased their political efficacy. Influence was governed by how effectively politics were perceived. This research can increase the political engagement of Western Chinese university students who utilize social media and offer some suggestions for how the government might carry out its daily operations to better control this activity.

**Keywords:** university students in Western China; social media use; online political participation; political efficacy





## 1. Introduction

On 4 May 1919, college students in China banded together at Tiananmen Square to demand the right to demonstrate at the Paris Peace Conference against China's unfair treatment. Members of the Harvard Student Labor Action Movement, who spoke out against exploitation and tyranny and pushed for a socialist future, organized numerous protests. From above, it is clear that college students are a vital force for political participation and that they have a unique role to play at crucial historical junctures in the development of society.

Social media growth has exploded as a result of the swift development of technology and the widespread usage of mobile phones. Social media has gradually assimilated into young people's daily lives and has developed into the main way for young people, particularly college students, to exchange experiences, ideas, and information. These distinctive technical qualities of social media enable robust interaction, practical use, open resources, and sharing at any time and any place. Chinese university students are increasingly posting, commenting on, and sharing information about contentious political topics such as the Sino–US trade war and the 40th anniversary of reform and opening up on social media platforms such as WeChat, Weibo, and QQ. In this way, using social media gives college students a very useful opportunity to participate in online politics.

The Western colleges are situated in a distinctive physical region of the country's interior that is multiethnic, and their political engagement is highly convoluted. Additionally, their political and cultural development is considerably slower than that of the eastern region. It is important to comprehend how university students participate in online political action since they have a huge impact on social development yet have information receiving channels that are more restricted when compared to university students in the east. This study, which uses empirical research methods, aims to investigate in depth the effects of social media use on university students in Western China. Additionally, it is important to discuss how it affects students' political efficacy in Western institutions as well as how it affects political participation online, and to more precisely target social media's positive effects on online political involvement among Western Chinese university students.

## 2. Literature Review

### 2.1. Political Participation

When it comes to political participation, various academics have varying perspectives. The current academic perspectives on whether or not political involvement is legal are extremely divergent.

Li (2001) defined political involvement as "the activities of ordinary citizens influencing the political power system in a certain way and the political behavior of important public political life" in the book "Modern Politics". Verba and Nie (1972) defined "political participation" as the legal actions taken by regular people to directly appeal to influence the decisions and actions of political and public officials, election campaigns, voting actions, etc.; various noninstitutional and illegal political actions are not considered to be a part of political participation. Nie's book "U.S. Participation: Political Democracy and Social Equality" was published in the 1970s. "Encyclopedia of China Political Science" (1992) defines political participation as "the behavior of citizens voluntarily participating in political life through various legal means". Wang (1995) holds that "political participation is ordinary citizens participating in political life through various legal methods and affecting the composition, the mode of operation, the rules of operation, and the behavior of the policy process of the political system".

Scholars such as Samuel P. Huntington (1989) elaborated on this idea by defining "political involvement" as embracing both autonomous action that influences government decisions as well as behavior that is influenced by others in order to influence government decisions. Political engagement can also take place through unlawful actions, such as violence (that is, political participation that violates present rules and regulations), in addition to legitimate actions such as election campaigns, voting actions, community activities, and special contacts. In recent years, an increasing number of academics have come to the conclusion that political engagement should cover all actions that have an impact on the political system of the government. Political participation, for instance, is described by Li and Liu (1995) as "the conduct of citizens trying to influence the political process through certain methods and channels".

Social media's introduction has further broadened the definition of political engagement by allowing individuals to express themselves and take part in political activities online, frequently outside of formal institutions. These activities range from founding and joining online groups to collaborating with friends online and creating self-organized online signatures. There are numerous ways to categorize different political activities at the moment.

Political participation is crucial in contemporary politics. On the one hand, it is the political framework that ensures the sound, steady, and long-term growth of a contemporary society; on the other hand, it is a crucial means of defending the political rights of the populace. Therefore, it is crucial to research how social media platforms actually influence people's political participation.

### 2.2. Political Efficacy

According to social cognitive theory, self-efficacy is one of the fundamental components of political efficacy. Self-efficacy holds that a person's self-assurance in their own beliefs, knowledge, or experiences improves behaviors. Successful outcomes of these acts lead to a high sense of self-efficacy, which aids in the prediction of an individual's future behaviors, reinforcing the degree of one's beliefs (Bandura 1997).

Similar to this, political science conceptualizes political efficacy as self-efficacy. An important indicator of whether individuals are motivated to participate in various types of political life has been the notion of political efficacy (Niemi et al. 1991; Finkel 1987). Political efficacy was initially described as "the belief that social and political change is both attainable and that each individual citizen may contribute to its realization" (Campbell et al. 1954). A person with high political efficacy exhibits active political views and behaviors that demonstrate a desire to engage in political systems. Because it guarantees the durability of participation, a high level of efficacy among citizens is seen as desirable from a democratic standpoint (Wright 1976). Studies also demonstrate that political efficacy is bolstered by diverse civic education or political experiences over the course of a person's life and is essential for enabling future democratic engagement (Kahne and Westheimer 2006; Mosher 1994).

More research has shown that political efficacy is divided into two categories: internal and outward political efficacy (Acock et al. 1985; Balch 1974; Niemi et al. 1991). Internal political efficacy is the belief or assurance that a person can comprehend politics or policies. A person's perception of how well public institutions or governments respond to their wants and desires is known as external political efficacy. The majority of research has found that, despite the two concepts' strong correlation, internal political efficacy is more stable since it reflects personal political characteristics, whereas external political efficacy is more unstable due to political experiences and situations (Kahne and Westheimer 2006). According to empirical research, internal efficacy is positively correlated with political activity, although external efficacy is not always in line with such activities because of its competing impacts (Spears 1999; Shingles 1981; Wollman and Stouder 1991).

In this study, political efficacy is referred to as internal political efficacy since it is challenging to quantify exterior political efficacy and because external political efficacy is difficult to manage due to the multiplicity of elements that can influence it.

### 2.3. Social Media and Political Participation of University Students

University students use the Internet very frequently and are very enthusiastic about using social media. They continually reinvent traditional culture and engage in a variety of social media activities with a strong sense of inventiveness. They create their own network culture, value system, and linguistic system during this invention process, all of which were fostered by the network culture. Since the Internet ecology has influenced young people's socialization processes since they were children, a large body of research demonstrates that young people are the most ready to use the Internet for interactions (Effing et al. 2011). Therefore, will college students who are more involved in a variety of social and political issues spark more interest in engaging in online political engagements or not?

Because of the social media's (including WeChat, Weibo, and QQ) explosive growth and the advantages of their immediacy, accessibility, and engagement, an increasing number of college students are opting to engage in politics online. Because the information on social media exhibits large, decentralized, and diversified qualities, university students are more likely to come into contact with current affairs and political information in the age of the Internet's information explosion than ever before. Whether actively seeking out information or passively absorbing it, university students can always come into conscious or unconscious contact with current affairs hotspot information in social media environments, such as perusing Tencent news, Weibo hotspots, and official accounts. The variety of political engagement opportunities for college students has increased, as has the bar for political engagement. Through social media, anybody may participate in politics, which not

only develops noninstitutionalized forms of political engagement but also institutionalized ones as well. In such a situation, university students' political efficacy is likely to amplify the favorable impact of social media on their political participation. The rise in political efficacy among college students is extremely likely to encourage students to browse and research the most recent political headlines as well as actively engage in communication and interactions on social media. When people come across significant political hot events, they could also decide to share it on social media or in the cyberspace, which will start conversations among friends or Internet users.

Social science research has paid a lot of attention to the connection between Internet use and political engagement. The contention that the web's selective nature rendered it unsuited for informing and mobilizing disinterested citizens predominated the early debates (Bimber and Davis 2003; Margolis and Resnick 2000).

### 3. Research Questions and Hypotheses

More students from the West, more students from rural areas, and more students who identify as members of racial and ethnic minorities are the main features that colleges and universities in the West portray. This is caused by elements including geographic location, social environment, and cultural heritage. Dear students, this is the fundamental difference between college students in Western China and those in Eastern China. These characteristics limit Western Chinese college students' ability to participate in politics to some extent.

Political communication has long centered on the study of how social media affects people's political participation. According to the study conducted thus far, there are three main schools of thought regarding the internal relationship between social media and political participation: "optimist", "pessimist", and "situational determinism". The "optimists" argue that social media usage encourages institutional and noninstitutional public participation in politics and plays an important role in doing so. The "pessimists" are dubious about "the beneficial impact of social media on political engagement". Pessimists point out that social media and other network applications either have no discernible impact or even harm users' political engagement. According to "situational determinism", the influence of social media on users' political engagement is multifaceted and subject to change as a result of adjustments to other factors.

Various academics have varying views on social media use and politics. Online social network use accounted for 5.8% of young people's likelihood of voting, according to Kim and Geidner (2008). According to Valenzuela et al. (2009), Facebook use intensity positively predicted civic engagement, whereas Facebook group use intensity increased both civic and political participation. According to Baumgartner and Morris (2010), using social networking sites (SNS) was associated with three online political behaviors: posting political statements on blogs, signing online petitions, and sending political emails or links. The use of social networking sites (SNS) and blogs was recently found to be one of the strongest predictors of online political activity in Kim and Chen's (2012) analysis of the 2008 Pew survey data. Gainous et al. (2013) examined the 2008 Pew survey data and came to the same conclusion that online social networking may encourage online political involvement. Furthermore, according to multiple national studies (Allstate/National Journal Heartland Monitor Poll XIII 2012; Rainie et al. 2012), the majority of social media users were politically and civically active individuals. Additionally, studies conducted outside of the US discovered that social networking usage predicted traditional active political participation (e.g., Bakker and De Vreese 2011). We may learn that social media use is related to civic engagement and political participation from the most recent literature on social media and politics. Based on that, the following research hypotheses are made:

**H1.** *The use of social media by university students at Western universities has a beneficial impact on their online political activity.*

Numerous studies have shown that political efficacy significantly and directly increases citizens' political participation. Political efficacy is typically viewed as one of the psychological aspects that influence political engagement the most. Political engagement is more enthusiastic when political efficacy is higher (especially when internal political efficacy is higher); when political efficacy is lower, political engagement is less motivated.

Social media users will be more motivated to debate politics on SNS, follow political figures on Twitter, and tweet or retweet about a political issue than other users if they feel they have political expertise and can affect the political process. Previous research demonstrated that political self-efficacy predicted the usage of conventional media for political objectives, such as call-in political television shows and talk radio (Newhagen 1994; Hollander 1996). Although social media currently lacks a wealth of empirical evidence, a few studies have found a link between political self-efficacy and blogging (Kaye 2005) and SNS use (Kim and Geidner 2008).

According to research, informed and self-assured people have a tendency to be politically engaged offline. Therefore, they should be more inclined than others to participate in political campaigns and other political activities online. In fact, a few recent studies (Jung et al. 2011; Gil de Zúñiga et al. 2012) found a favorable correlation between political self-efficacy and online political participation. As a result, we surmise that:

**H2.** *Political efficacy has a positive effect on the online political participation of university students in Western universities.*

Political efficacy is a term used to describe how much a person thinks they can affect the political system. Political efficacy and a number of participation behaviors have a positive correlation, according to decades of research and measurements on the topic. However, due to their lack of formal learning opportunities in school and university and their apparent incapacity to engage in politics and public affairs, young adults' "political experiences are typically left to happenstance and the influence of individual backgrounds" (p. 554) because the media offers a symbolic setting that enables young people to " . . . process and transform transient experiences into cognitive models that serve as guides for judgment and action" (p. 267). The relationship between Internet use and political efficacy has been previously explained in both Chinese and American contexts. SNS are especially well suited to support experiential learning and encourage reinforcement for individuals who are politically effective because of their media-rich and interactive characteristics.

According to the justification given above, people who already have high levels of political efficacy will have a stronger association between using the Internet and social media and participating in activities. It is also feasible, particularly in the context of youth, that the association will be stronger for people who have lower levels of political efficacy. This is due to the fact that young adults with higher levels of political efficacy often have more civic resources, such as learning and participation opportunities supplied by their families or organizational affiliations, that are independent of social media use. However, those with relatively little resources are required to use social media to engage in politics and public affairs. Young individuals with lower levels of political efficacy can make up for the lack of political and civic learning opportunities accessible in their daily lives by using social media. Therefore, we suggested a different research design:

**H3.** *The beneficial effects of social media use on university students' online political participation in Western China are strengthened by political efficacy.*

## 4. Research Method

This research uses questionnaires to explore the possible relationship between the use of social media, political efficacy, and political participation of university students in Western China. The subjects of the research are university students between the ages of 18–40 in Western China. In total, 320 copies of the questionnaire were distributed online through the "investigation group" (http://www.diaochapai.com, accessed on 16 March 2022), and

242 copies of field visits to offline universities were issued. A total of 564 questionnaires were issued, and 530 valid questionnaires were recovered, with an effective response rate of 93.97%. The title of the questionnaire is "Research on the Impact of University Students' Social Media Use on Online Political Participation". The online snowball method was used for questionnaire distribution. The questionnaire was distributed through social media sites such as WeChat groups, QQ groups, and Weibo to university students in Western China who had different educational backgrounds, nationalities, majors, political identities, and family economic statuses. The specific method was as follows: First, randomly select 30 students from different universities with different education backgrounds, majors, and genders in Western China through social media as the first group of survey subjects, and then invite them to use their social media to invite their friends and classmates from universities in Western China to fill out the questionnaire. The questionnaire was distributed from 10 June 2019 to 20 June 2019, and 320 people filled out the questionnaire. After removing invalid questionnaires such as those where the answer time was too short and the same options were chosen for all questions, 302 valid questionnaires were returned online, and the effective response rate was 94.38%. Offline random sampling was used to distribute field questionnaires at Guangxi University, Nanning Normal University, Yunnan University, Guizhou University, Guangxi Normal University, and other universities that were located in Western China. A total of 244 questionnaires were distributed offline, and 228 valid questionnaires were recovered. The effective recovery rate was 93.44%.

In this study, the in-depth interviews and surveys complement one another in their use. The results of the questionnaire analysis were used to draw the conclusions of the pertinent hypotheses, and the in-depth interviews further confirmed the validity and reliability of the data analysis based on those findings.

SPSS23.0 and Stata13.0 were used to analyze the questionnaire data. (1) SPSS23.0 data processing software was used to test and analyze the reliability and validity of the questionnaire. Additionally, we use it to perform descriptive analysis and correlation analysis on the pertinent questionnaire variables. (2) We used Stata13.0 data processing software to analyze the correlation between the main variables and grasp whether there was a correlation between the variables and the degree of correlation. We simultaneously carried out multiple regression linear analysis on the control, independent, and dependent variables. Additionally, we performed White's test to determine whether the moderating variable was heteroscedastic before making a change. At last, some contents of the questionnaire were supplemented in the form of in-depth interviews. The in-depth interviews were conducted by recruiting interview volunteers in the universities. A total of 10 university students in Western China participated in the in-depth interviews: 5 male university students and 5 female university students. The educational backgrounds ranged from an undergraduate to master's to doctoral degree. The interview was conducted in a face-to-face semi-structured interview mode, and the interview time was controlled within 50 min.

### 4.1. Dependent Variable

The dependent variable of this research is online political participation, that is, political participation through social media. The online political participation scale refers to the research of Zuniga and Ozman and uses 11 questions to measure online political participation. Respondents were asked to indicate the frequency of their participation in these activities using a Likert five-level scale (1 = never, 2 = occasionally, 3 = less, 4 = often, 5 = frequently). By accumulating the scores of these 11 questions, we obtained the value of the online political participation of university students in Western China.

From the frequency of participation, we found that the higher frequency of online political participation activities of university students in Western China was mainly concentrated in the following ways: browsing current political news or political hot events on social media or websites; visiting governments websites at all levels or following their official social media accounts; visiting the website of public administration departments

(communities, schools, companies, etc.) or following their official social media accounts; and participating in online survey activities. On the other hand, the relatively low frequency of online political participation of university students in Western China included protesting against a decision or behavior of the government or public administration department through the Internet; contacting the government department or the leader of a certain unit through a private message on the Internet; and participating in online signature petition activities (such as candle lighting, etc.) or online rights protection.

Using the pertinent information and a thorough examination of the in-depth interviews, We discovered that university students in Western China participate in gathering political information on a somewhat regular basis through viewing the news on television, reading the news in newspapers, and browsing current political hot events online. Social media has facilitated online political participation for university students in Western China.

However, some interviewees stated that they would not actively search for current affairs news. Among the 10 interviewees, 4 people said that they mostly receive news hotspots automatically pushed by social media sites such as WeChat and QQ and said "I mainly click on news feeds casually, holding a mental attitude of taking a look at it" (P3). The news pushed on social media will not be searched and understood in depth, unless you encounter news related to your own interests. Through interviews, it was found that "visiting the website of public administration departments (communities, schools, companies, etc.) or following their official social media accounts" and "social current political news or political hot events on social media or websites" are highly active online political participation behaviors of university students in Western China. Six of the ten interviewees actively looked for news across multiple channels.

We have discovered that university students in Western China participate in political action on a rather infrequent basis. For example, "protesting against a decision or behavior of the government or public administration department through the Internet" or "contacting the government department or the leader of a certain institute through a private message on the Internet". Respondents regarded the above behaviors as high-risk behaviors, so they adopted a more conservative attitude. The interviewee (P8) said, "Even if I make comments on relevant government policies, do officials have time to read it? Can it be solved if they saw it?" The interviewee (P1) who reported the problem to the principal through the principal's webmail did not feel that he had achieved the desired effect by writing to the principal. He believed that the reply was not necessarily the principal personally replying, and the reply content was also very official.

The interview also found that the frequency of forwarding and online communication is not high, that is to say, statements such as "forward or publish articles or posts about social current political news or political hot events;", "discussing current political topics with netizens or classmates through the Internet", etc. received low scores in questionnaire. One interviewee (P5) said "The Internet is a place where people are mixed, and there are all kinds of people. If some words are spoken, they will be besieged by some netizens. Some questions become more and more confusing. It's better not to talk about it and save yourself from the trouble". Another interviewee (P6) said "Even if you make a comment, no one will notice it, let alone promote social progress". This shows that the online political participation of university students in Western China is mainly focused on the participation in browsing and commentary and is not very enthusiastic about action-style political participation.

Regarding the causes of the above phenomenon, on the one hand, due to the emergence of social media, the frequency of pushing currently hot political information has increased. Even if it is passively received, it will also increase the number of times university students view and browse such information. For currently hot political news of the day, university students in Western China can browse through WeChat Moments, Weibo topics, and Qzone multiple times; on the other hand, some university students in Western China believe that action-style political participation is riskier and more difficult, so they will adopt a more cautious attitude and will not take action easily.

*4.2. Independent Variables*

The influence of social media is gradually increasing among university students. More and more university students choose to browse news, communicate, and go shopping through social media. In order to distinguish social media use from general Internet use and facilitate data analysis, this research introduces general Internet usage variables in the design of the questionnaire variables and puts it with social media and political efficacy as a whole variable module.

Our survey makes use of the Tom P. Bakker and Claes de Vreese Internet Usage Scale. Based on the settings of two questions, we determined how frequently the Internet is used for general use (not including social media use). Respondents were asked to use a Likert 6-level scale and a Likert 7-level scale (1 day–7 days) to indicate their average usage time per week. The product of these two questions is the assignment of the duration of general Internet use.

Social media is a typical application based on Web2.0 technology. It is a tool and platform on which users can produce content and communicate with others. Social media is a new type of online media that gives users a high sense of participation and a huge cyberspace. It has basic characteristics such as participation, sharing, communication, and connectivity. Weibo, WeChat, and QQ are the most popular social media platforms among Chinese Internet users at the moment. We used the general Internet usage scale above to measure the usage of Weibo, WeChat, and QQ, and then we added the values of Weibo, WeChat, and QQ use to obtain the social media variable value.

As for political efficacy, the questionnaire in this research used a scale that was designed by Richard Niemi, Stephen Craig, and John Gastil. Respondents needed to choose their degree of approval of the following questions on the Likert five-level scale (1 = completely agree, 5 = strongly opposed). The sum of the scores of these four items was the political efficacy value of the university students in Western China. The higher the score, the higher the political efficacy, and vice versa. According to the data analysis, the overall degree of the political efficacy of university students in Western China was not low, which indicates that university students in Western China have confidence in their ability to participate in politics.

*4.3. Controlled Variables*

This study segmented the controlled variables into modules of demographic variables, prepolitical variables, and social network scale variables to minimize the design error (10 variables in total). The six variables in the demographic variable module were age, gender, ethnicity, major, household income, and academic qualification. Open-ended questions were used to directly measure age (M = 23.51 SD = 3.20) and gender (M = 0.47 SD = 0.50), with 1 denoting a male and 0 denoting a female. For Han nationality, the ethnic variable was 1, while for minority, it was 0. A value of 1 stood for the humanities and social sciences in the primary variable, and 0 for the natural sciences and other fields. The interviewees' average monthly income (1 = below 2000, 2 = 2000–4000, 3 = 4000–8000, 4 = 8000–15,000, and 5 = over 15,000) (M = 3.25 SD = 1.27) was used to measure the economic income variable. Regarding the academic qualification variable, 1 corresponded to undergraduate students and 2 to graduate students and above (M = 1.46 SD = 0.50). The prepolitical variable module was made up of political identification, political interest, and political orientation. The CCP members and the Communist Youth League members were represented by 1, and the masses and democratic parties were represented by 0 (M = 0.89 SD = 0.31). Political identity was measured using multiple-choice questions, with the options being the masses, the Communist Youth League members, the CCP members, and the CCP. A five-level Likert scale was used to gauge political interest. In most cases, we asked the respondent "Are you interested in hot social current events" (1 = not interested at all, 2 = not interested, 3 = neutral, 4 = interested, 5 = very interested) and "Your degree of agreement with Politics is very interesting" (1 = resolutely disagree, 2 = disagree, 3 = neutral, 4 = agree, 5 = completely agree), and then we added the scores of the two items to determine

the value of political interest, whereby political interest increases and decreases inversely with the score. The Likert five-level scale was used to gauge the interviewees' political orientations and their level of agreement with the following viewpoints (1 = strongly agree, 2 = agree, 3 = neutral, 4 = disagree, and 5 = vehemently disagree): (1) the CCP and government should control everything that happens in the nation; (2) they should choose the ideology or notion that will be promoted across society; and (3) diverse viewpoints will cause social unrest. The value of political orientation (M = 9.84 SD = 2.23) is the total of these three items' values. More open mindedness is demonstrated by higher scores and vice versa. When asked "How many friends are there in your most popular social media accounts", the respondent's response was utilized to calculate the social network scale variable module (M = 3.46 SD = 1.05). Less than 50 people was represented by 1, between 50 and 100 by 2, between 100 and 200 by 3, between 200 and 500 by 4, and beyond 500 by 5. The size of the social network increased as this item's score increased.

### 4.4. Reliability and Validity Testing

In terms of reliability testing, this research sorted out and analyzed the questionnaire data and found that the Cronbach's Alpha coefficients of the main variables were as follows: political efficacy was 0.819, social media use was 0.780, online political participation was 0.873, and all of them fit the reliability requirements. The overall reliability of the questionnaire was also tested. The Cronbach's Alpha coefficient was 0.849, which is greater than 0.8, indicating that the reliability was very good.

In terms of validity testing, we strictly followed the standard preparation procedure to design the questionnaire, which basically guaranteed the validity of the questionnaire. We used the SPSS23.0 software to perform the KMO and Bartlett test on the questionnaire. The overall KMO value was 0.873, which is greater than 0.6. The Bartlett's sphericity test's significance was $p < 0.001$, indicating that the questionnaire's construct validity was strong overall.

## 5. Results

### 5.1. Descriptive Statistical Analysis of Sample Data

We gathered data and put together valid questionnaires, and we discovered that the distribution of degrees, majors, and gender was generally balanced. Age, ethnicity, and political identity, however, were all distinct. The youngest undergraduate student was 18 years old, and the oldest PhD student was 39 years old. The average age was 23.51. The age varied from 18 to 40, and in terms of ethnicity, Han nationality accounted for 82.08% of the population while minorities accounted for 17.92%. The questionnaire was disseminated across Western China, where there are minorities including the Miao and the Buyi, with the Zhuang being the largest. In terms of political affiliation, 53.77% of people were Communist Youth League members. Members of the nonparty league made up 10.76% of the total population, while CCP members made up 35.47%. According to 27.36% of respondents, their families' monthly incomes fell between $4000 and $8000. The next highest monthly income range was between 8000 and 15,000, accounting for 24.15% of respondents, and the lowest was between 2000 and 8000, only accounting for 11.51% of respondents. In total, 37.17% of respondents had 200 to 500 friends, which was the highest proportion according to social network size metrics. The second, accounting for 28.11% of respondents, was the group of 100 to 200 individuals. Only 3.78% of respondents had fewer than 50 friends, which was the lowest percentage. Table 1 displays the descriptive statistics for the primary controlled variables.

**Table 1.** The descriptive statistics of the main controlled variables.

| Variable | Sample Size | Mean (Standard Deviation) | Variable Description |
|---|---|---|---|
| Gender | 530 | dichotomous variable | 0 = Female (53.4%), 1 = Male (46.6%) |
| Nationality | 530 | dichotomous variable | 0 = Minority (17.92%), 1 = Han nationality (82.08%) |
| Major | 530 | dichotomous variable | 0 = natural sciences and other majors (50.19%), 1 = humanities and social sciences (49.81%) |
| Average Monthly Income | 530 | 3.25 (1.27) | 1 = Below 2000 ¥ (11.51%), 2 = 2000 ¥ to 4000 ¥ (16.79%), 3 = 4000 ¥ to 8000 ¥ (27.36), 4 = 8000 ¥ to 15,000 ¥ (24.15%), 5 = More than 15,000 ¥ (20.19%) |
| Academic Qualifications | 530 | dichotomous variable | 1 = undergraduate (54.34%), 2 = master's degree and above (45.66%) |
| Political Identity | 530 | dichotomous variable | 0 = The Masses and Democratic Parties (10.76%), 1 = Members of the CCP and the Communist Youth League (89.24%) |
| Social Network Scale | 530 | 3.46 (1.05) | 1 = Less than 50 people (3.78%), 2 = 50 to 100 people (15.28%), 3 = 100 to 200 people (28.11%), 4 = 200 to 500 people (37.17%), 5 = More than 500 people (15.66%) |

*5.2. Research Findings*

Regression analysis is based and premised on correlation analysis. Through the analysis of the Pearson correlation coefficients of the main variables in the research questionnaire, it was found that there was a significant positive correlation between the social media use of university students in Western China and online political participation. Social media use was substantially connected with online political activity among college students in western China ($p < 0.001$), which indicates a positive effect ($\beta = 0.051$).

Therefore, it can be explained that the longer social media is used by university students in Western China, the higher their online political participation. Like the positive promotion effect of social media use, the political efficacy of university students in Western China had a significant positive correlation with their online political participation. Western Chinese college students' online political engagement is positively connected with political efficacy ($\beta = 0.323$) and significantly correlated with it ($p < 0.001$). It can be shown that the higher the political efficacy of university students in Western China, the higher their online political participation. The results obtained through the regression analysis were basically consistent with that obtained with the correlation analysis. Table 2 shows the results.

According to the results of the regression equation, there was a significant positive correlation between the online political participation of university students in Western China and the interactive variables (political efficacy social media use) with regard to the moderating effect of political efficacy. The university students in Western China who participate in politics online significantly ($p < 0.05$) and favorably ($\beta = 0.005$) affected the interactive item (political efficacy social media use). This demonstrates that social media

use has a favorable impact on university students' online political participation and vice versa, inversely proportional to their level of political efficacy.

**Table 2.** Regression analysis table of the influence of social media use on online political participation of university students in Western China.

| Predictor Variable | Online Political Participation | | | |
|---|---|---|---|---|
| | M1 | M2 | M3 | M4 |
| Age | −0.272 ** (−2.31) | −0.332 *** (−3.02) | −0.284 *** (−2.62) | −0.289 *** (−2.68) |
| Nationality (Han Nationality = 1) | −0.713 (−0.86) | −0.895 (−1.17) | −0.418 (−0.55) | −0.295 (−0.39) |
| Gender (Male = 1) | 2.670 *** (4.05) | 1.580 ** (2.58) | 1.640 *** (2.73) | 1.613 *** (2.69) |
| Major (humanities and social sciences = 1) | 1.739 *** (2.63) | 1.210 ** (1.97) | 1.063 * (1.76) | 1.088 * (1.81) |
| Academic Qualification (postgraduate and above = 1) | 2.733 *** (3.61) | 1.662 ** (2.36) | 1.391 ** (2.00) | 1.379 ** (1.99) |
| Income | 0.184 (0.74) | −0.037 (−0.16) | 0.085 (0.36) | 0.099 (0.42) |
| Adjust $R^2$ (%) | 5.00 | | | |
| Political Identity (party member = 1) | | −0.558 (−0.57) | −0.570 (−0.59) | −0.571 (−0.59) |
| Political Interest | | 1.567 *** (6.40) | 1.457 *** (6.02) | 1.490 *** (6.15) |
| Political Orientation | | 0.388 *** (2.73) | 0.283 ** (1.97) | 0.286 ** (1.99) |
| Social network scale | | 1.449 *** (4.98) | 1.261 *** (4.35) | 1.263 *** (4.36) |
| Adjust $R^2$ (%) | | 20.84 | | |
| Use of Internet | | | −0.054 (−1.58) | −0.056 * (−1.65) |
| Use of Social Media | | | 0.051 *** (3.42) | −0.013 (−0.31) |
| Political Efficacy | | | 0.323 *** (3.62) | 0.019 (0.09) |
| Adjust $R^2$ (%) | | | 23.89 | |
| Political Efficacy × Use of Social Media | | | | 0.005 * (1.66) |
| Adjust $R^2$ (%) | | | | 24.15 |

* $p < 0.05$, ** $p < 0.01$, *** $p < 0.001$. − in the table is the standardized regression coefficient, show as β. The t value is in brackets.

We used political efficacy as the moderating factor. The adjusted model effect in the output of process had a 95% confidence interval of [0.001, 0.021], excluding 0, showing that the general model's establishment was justified. The respondents' online political involvement behavior increased with their view of their political efficacy, and there was therefore always a positive correlation between the use of social media and online political engagement.

The age variable among the demographic indicators had a negative effect on online political involvement (β = −0.289), and the regression was significant ($p < 0.001$), which indicated that the younger the university students in Western China are, the higher the

online political engagement. Gender and online political participation ($\beta$ = 1.613) had a positive correlation in terms of gender factors, and the regression was significant ($p < 0.001$), indicating that male university students participate more in online politics than female university students. The major variables had a positive impact on online political participation ($\beta$ = 1.088), and the regression was significant ($p < 0.05$), demonstrating that the university students majoring in the humanities and social sciences participated in politics more than the students majoring in the natural sciences and other disciplines. Academic qualifications and online political activity had a positive association in terms of academic factors ($\beta$ = 1.379), and the regression was significant ($p < 0.01$). Higher educated university students therefore exhibited greater levels of online political activity. Online political engagement was significantly correlated with political interest ($p < 0.001$), and there was a significant positive link between political interest and online political participation. This shows that there is a direct correlation between political interest and online activity among university students in Western China. Online political involvement significantly increased political orientation ($p < 0.01$) in terms of factors related to political orientation, which is a good thing. This demonstrates that university students in Western China who are more politically engaged online are more open minded; according to the social network scale variable, there was a significant positive correlation ($\beta$ = 1.263) between the social network scale and online political participation, and the regression was significant ($p < 0.001$), which means that the more social network friends that university students in Western China have, the higher their level of online political participation. The three factors of ethnicity, economic status, and political identity had no discernible impact among the other controlled variables.

## 6. Conclusions

The following conclusions are taken from our study, which looked at and evaluated the effect of social media use on students' online political activity using university students in Western China as its research subjects:

First, Western Chinese university students who utilize social media are more likely to participate in online political discourse. The longer Chinese university students use social media, the more likely they are to obtain the most recent political news. There are numerous and instantaneous social media updates regarding hotspots in current events. The resonance effect on the same screen will enhance the perceptions of Western Chinese university students and provide them with access to a wide range of political data. Students in universities in Western China now have broader perspectives on politics thanks to the benefits of political literacy. Additionally, Western Chinese university students can engage in politics more effectively and practically through social media. On social media, students can share and comment on trending current events topics whenever and from wherever they are. Additionally, they can disseminate news about current events among groups connected on social media. This can spark further discussions with close relatives, friends, or online users. In some cases, online discussions also influence offline conversations and behaviors.

The more frequently someone utilizes social media, the more actively they engage in politics online, according to what was found with university students in Western China. First of all, university students have a tendency to be more outgoing, adaptable, and willing to try new things. On social media, university students can discuss recent current events hotspots and offer their perspectives on these events. They can also comment on matters to the heads of relevant departments via emails or private messaging. Due to the growth of social media, which provides the benefits of instant satisfaction, interaction, and simplicity, university students now have new opportunities and platforms to participate in online politics. Second, university students may gain likes or comments from Internet users if they post original opinions on social media or share politically relevant breaking news. Online political participation, positivity, and zeal among university students are increased by one-click sharing, likes, and other social media features. Not the least among other things,

social media gives college students the chance to learn a variety of varied information. Additionally, it increases their chance of being exposed to political information, which can kindle and foster an interest in politics. In this scenario, there will unavoidably be an increase in online political engagement.

Second, as a result of their greater political efficacy, university students in Western China will become more politically engaged online. This shows that the more effective they are politically, the more actively Western Chinese university students engage in politics online. When college students feel increasingly satisfied with the political climate and believe they can have a greater influence on the process, their involvement, passion, and motivation for online political activity will increase. The results indicate that this effect is extremely considerable, illuminating the significance of political efficacy for online political participation among university students. On the other hand, university students will have greater direct access to political information as political efficacy increases. They will have a better understanding of political structures and guiding ideologies as a result, which will motivate them to participate in political activities online.

Third, political efficacy can enhance the beneficial function that social media use can perform in encouraging online political involvement. Accordingly, the greater the favorable effect of Western Chinese university students' usage of social media on online political participation, the more effective they will be politically. On the one hand, the more effective a university student is politically, the better the environment is for fostering a stronger sense of political identity and increasing their satisfaction with the political environment, which in turn increases their confidence and willingness to engage in online political activity through social media. On the other hand, if a person's political effectiveness rises, they are more likely to actively share and discuss current political information through social media or join pertinent interest groups, which raises their level of online political participation.

Fourth, the degree to which university students in Western China engage in politics online is influenced by factors that can be controlled, such as gender, major, political interest, and social network size. Male college students engage in politics online to a greater level than female students. University students majoring in humanities and social sciences participate in politics to a greater extent than students majoring in natural sciences and other subjects. Additionally, political involvement online is increased among university students in Western China who are more interested in politics and among those who have larger social networks.

The impact that social media use has on college students' online political participation cannot be understated. Due to the rapid growth of social media, Internet users are talking more and more about social and public events. In order to express themselves and forward, comment on, and like the news stories that interest them or that they agree with, more and more university student organizations are turning to social media. They also freely participate in conversations about politics and public policy. A critical period of social transition is currently occurring in China. Therefore, it is important to research how western Chinese college students utilize social media to engage in online politics.

There is still potential for improvement in the measurement of various concepts during the questionnaire development procedure in this study. For instance, when measuring political efficacy, we ignored the differences between introverted and extroverted efficacy and lumped them together into one category. When analyzing the impact of Weibo, WeChat, and QQ use, we did not conduct a horizontal comparative analysis of the three platforms' effects on political engagement, which could have impacted the sample heterogeneity. Future research can solve these flaws and restrictions and continue the in-depth investigation.

**Author Contributions:** Conceptualization, Y.T. and Q.W.; methodology, Y.T. and Q.W.; software, Y.T.; validation, Y.T.; formal analysis, Y.T. and Q.W.; investigation, Y.T. and Q.W.; resources, Y.T. and Q.W.; data curation, Q.W.; writing—original draft preparation, Y.T. and Q.W.; writing—review and editing, Q.W; visualization, Y.T.; supervision, Q.W.; project administration, Q.W. All authors have read and agreed to the published version of the manuscript.

**Funding:** The essay was funded by the Project: Beijing Institute of Graphic Communication's doctoral start-up: Research on the Influence Mechanism of College Students' Privacy Protection in Social Media Environment, which was funded by Beijing Institute of Graphic Communication.

**Informed Consent Statement:** Informed consent was obtained from all subjects involved in the study.

**Data Availability Statement:** The data presented in this study are available upon request from the corresponding author. The data are not publicly available to protect the privacy of the interviewees and participants.

**Conflicts of Interest:** The authors declare no conflict of interest.

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
