# Peer review of "An Empirical Study of the Impact of Social Media Use on Online Political Participation of University Students in Western China"

_journalmedia, doi:10.3390/journalmedia4010006_

Round 1

Reviewer 1 Report

-Please re-structure the paper as you have written materials and methods at the 4th point ideally it should be the 2nd point.

-The tables for Factorial Analysis must be given.

-You have only proposed the Hypotheses in the Discussion where is mention of these hypotheses in the result, have you done the testing? what is the result of those hypotheses?

Author Response

Q1: Please re-structure the paper as you have written materials and methods at the 4th point ideally it should be the 2nd point.

Response: The article's various sections have already been rearranged so that it may be presented logically from the introduction through the literature review, hypotheses, methodology, and results to conclusions.

Q2: The tables for Factorial Analysis must be given.

Response: We have borrowed mature foreign scales, fine-tuned them according to China's actual national conditions, and tested the overall reliability and validity, hoping that this can meet expectations.

Q3: You have only proposed the Hypotheses in the Discussion where is mention of these hypotheses in the result, have you done the testing? what is the result of those hypotheses?

Response: Actually we have done the hypotheses testing before we conduct the survey, while we do not show it detailed in the article. As Follows:

The questionnaire data were sorted and examined for this study's reliability testing, and it was found that the Cronbach's Alpha coefficients for the study's key variables are as follows: Online political involvement, social media use, and political efficacy all meet the criteria for reliability. The overall Cronbach's Alpha coefficient for the questionnaire is 0.849, which is better than 0.8 and indicates extremely strong reliability, according to the reliability test.

In order to assess the validity of the study, it thoroughly sorted out the ideas of social media usage, political efficacy, and political engagement among college students. It also made some initial adjustments to the questionnaire and asked friends to fill it out. In order to more fully comprehend the respondents' comprehension and cognition of each questionnaire item, it was necessary to undertake targeted questionnaire adjustments before forming the study's final questionnaire. The validity of the questionnaire's content was essentially assured throughout the process by rigorous adherence to the standard preparation technique. The questionnaire was subjected to KMO and Bartlett tests using SPSS 23.0 software. The questionnaire's overall KMO score was 0.873, which is higher than 0.6. The questionnaire as a whole has a fair amount of structural validity, as evidenced by the Bartlett test's significance p value of 0.000 and the result being less than 0.001.

Reviewer 2 Report

Pls. kindly refer to the attachment for the comments and suggestions.

Author Response

Q1: The structure of the whole article needs to be reorganized carefully. The presentation order of the background, literature, hypotheses, methods, results, and discussions should be revised.

Response: The article's various sections have already been rearranged so that it may be presented logically from the introduction through the literature review, hypotheses, methodology, and results to conclusions.

Q2: The moderating effect was not well supported. In line 221, the p value for the interaction term is ‘<.1’. The authors are suggested to explain the significance level they chose. A p value less than .1 is not strong evidence. Therefore, the role of political efficacy claimed by the authors could not be proved with confidence.

Response: Political efficacy was selected by the researchers as the moderating factor. The 95% confidence interval of the adjusted model effect in the output of Process is [0.001, 0.021], excluding 0, demonstrating that the establishment of the general model is supported. As a result, there is always a positive link between the usage of social media and online political engagement, and the respondents' online political participation behavior increases with their perception of their political efficacy.

Q3: The theoretical research questions and related hypotheses of the current study had been investigated intensively in past research. It seems that the biggest contribution of the study is that it focused on a group of young people in western China. The innovativeness of the study is therefore limited. What is the uniqueness of the case of university students in western China? What are the differences between university students in western and eastern China? The article was suggested to offer a more detailed description of the social, cultural, and educational background of university students in China’s western region. Also, a review of the political opportunity structure of western China’s society may help refine the research questions and hypotheses.

Response: We are aware that this study's innovation is somewhat constrained. The economic development of western China, however, is relatively behind that of the eastern region due to its geographical location, social environment, and cultural background deep in China's inland. This can be seen starting from the development characteristics, political environment, and social education background of western China. As a result, western China's colleges and universities tend to have a higher proportion of western, rural, and ethnic minority students, which triggers out that the university students in east China are more open-minded than the westerns. According to the related literature review, we can know that more studies were focus on the university students in east China, so we prefer to research the university students in west China, and both part of research formulate the university students in China in social media age how to be participated in political actions.

Q4: The literature review part of the article introduced and discussed definitions of the core concepts. A review of more relevant and recent studies is suggested to be carried out. Past research related to the theoretical relationships stated in hypotheses should be discussed more thoroughly and with more depth. The theoretical importance of the case of western China’s university students is expected to be addressed.

Response: This is a great suggestion. And we have already updated more literature review to support our research questions and hypotheses.

Q5: Research methods are supposed to be described more adequately. It appears that most of the samples were drawn from southwestern China. How about other regions of western China? The representativeness and external validity should be discussed. Measurements of the variables should also be presented in detail. Major of the students was measured as a dichotomous variable. The reason for such a way of measuring needs to be clarified. Why did the study control for political identity, political interest, political orientation, and social network scale? A few of the control variables are supposed to be discussed briefly. Furthermore, political interest and political orientation did not appear in Table 1. The article reported a few results from correlation analysis (lines 194-208), a correlation matrix table may be helpful for showing the results.

Response: As for the method part, we have updated it to make it more logical. As for the major of the students, which was measured as a dichotomous variable, which is its real existence in China, usually we collected the major from these two categories. From the literature review part, we know that the political interests will affect the political participation, while the political identity, political interest, political orientation, and social network scale are related to the political interests and thus will affect the political participation, so we control those variables.

Q6: The relationship between in-depth interviews and the questionnaire survey was not clearly explained. It seems that the in-depth interview served as a supplementary part for the quantitative findings. The rationale for the qualitative study is neither strong nor convincing.

Response: In-depth interviews and surveys complement one another in their use. The results of the questionnaire analysis were used to draw the conclusions of the pertinent hypotheses, and the in-depth interviews further confirmed the validity and reliability of the data analysis based on those findings. Combining the two distinct methodologies to improve the clarity of the research findings.

Q7: 7. In the research results part, descriptive statistics of core variables should be presented first. A simple slope analysis is supposed to be reported in the research findings.

Response: In the result part, we have already presented the descriptive statistics of core variables.

Q8: Some of the conclusions may need to be refined or further clarified. The article reported that ‘The participation frequency of university students in western China for action-style political participation is relatively low’ (lines 415-416). The article concluded that ‘sometimes online discussions also trigger offline discussion and actions’ (lines 512-513). This conclusion is very interesting and of theoretical importance. Under what conditions online discussion can trigger offline discussion and actions? Had the authors ever considered exploring public issue-related information receiving and other online participation behaviors separately? More complete and in-depth analyses and discussions are expected.

Response: Updated information has been made on the aforementioned misunderstandings. Due to their own worries about how engagement in action-style politics will affect their daily lives, jobs, confidence, and other factors, it is quite low. We have another article to convey the more in-depth debate, therefore in this one we only want to provide the data results. This should make sense, I hope.

Q9: Tables are suggested to be revised. In Table 1, the variable description for Average Monthly

Household Income was not accurate. In Table 2, R2 change values should be presented.

Response: "Average monthly household income" has been changed to "average monthly income." Table 2's Adjust R2 value is 25.14.

Q10: In line 494, the p value was ‘0.000’. This statement is incorrect.

Response: The Bartlett's sphericity test's significance was p<0.001, indicating that the questionnaire's construct validity was strong overall.

Round 2

Reviewer 1 Report

Thanks. 

Author Response

Dear Reviewer,

Really grateful for the slight revision opportunity. We are incredibly honored to resubmit our article. We value the helpful recommendations provided by the reviewers very highly. We carefully revised the material and followed all the suggestions made by the review panel. Different colors for additions are used to indicate the text modifications. Following are the responses to the reviewer's criticisms.

We hope the revised draft of the manuscript allayed your concerns, and we anxiously look forward to hearing from you.

Sincerely,

Qing WEN

Reviewer#1:

I hope it makes sense because the terminology has been altered and rearranged throughout the entire article.

Reviewer 2 Report

Thank you very much for your response and clarification. The revised manuscript has made a lot of improvements. The structure is clearer and the logic is more coherent. Some minor revisions are suggested to be made.

1. The so called political effectiveness in the abstract may cause confusions. Pls. make sure that this expression is correct. If it means political efficacy,  using political efficacy will be better.

2. H3 is suggested to be proposed more accurately. Is it trying to state the moderation effect? The hypothesis was not clear enough. 

3. Descriptions for the measurements of independent variables are supposed to be more complete, especially political interest and political orientation. 

4. The results are suggested to be described in a more precise way. In Table 2, the interactive term was significant at .05 level, but the line 469 said the interactive term was significant at .1 level, which one is the true result? Besides, “The online political participation of university students in western China has a significant impact on the interactive term”(line467-468), this sentence did not state the relationship between interactive term and dependent variable in an accurate way. Pls. recheck it and many other sentences to make sure that the research findings are described accurately.

Author Response

Dear Reviewer,

Really grateful for the slight revision opportunity. We are incredibly honored to resubmit our article. We value the helpful recommendations provided by the reviewers very highly. We carefully revised the material and followed all the suggestions made by the review panel. Different colors for additions are used to indicate the text modifications. Following are the responses to the reviewer's criticisms.

We hope the revised draft of the manuscript allayed your concerns, and we anxiously look forward to hearing from you.

Sincerely,

Qing WEN

Reviewer#2:

Q1: The so-called political effectiveness in the abstract may cause confusions. Pls. make sure that this expression is correct. If it means political efficacy, using political efficacy will be better.

Response: Political efficacy has been used in place of every instance of political effectiveness to make the essay more readable.

Q2: H3 is suggested to be proposed more accurately. Is it trying to state the moderation effect? The hypothesis was not clear enough.

Response: After some consideration, we altered the H3 to say: The beneficial effects of social media use on university students' online political participation in western China are strengthened by political efficacy.

Q3: Descriptions for the measurements of independent variables are supposed to be more complete, especially political interest and political orientation.

Response: We added Descriptions for the measurements of independent variables to increase the precision and clarity of the explanation of political interest and political orientation.

Q4: The results are suggested to be described in a more precise way. In Table 2, the interactive term was significant at .05 level, but the line 469 said the interactive term was significant at .1 level, which one is the true result? Besides, “The online political participation of university students in western China has a significant impact on the interactive term”(line467-468), this sentence did not state the relationship between interactive term and dependent variable in an accurate way. Pls. recheck it and many other sentences to make sure that the research findings are described accurately.

Response: Following a thorough evaluation, we made some corrections to the table 2 description. And to improve the article's accuracy, restructure the result section of the article.
